# Clinical Burden and Healthcare Utilization Associated with Hospitalizations of RSV-Infected Polish Children During the 2022/23 Season

**DOI:** 10.3390/v18010060

**Published:** 2025-12-30

**Authors:** Jan Mazela, Teresa Jackowska, Marcin Czech, Ewa Helwich, Oliver Martyn, Pawel Aleksiejuk, Anna Smaga, Andrzej Tkacz, Joanna Glazewska, Jacek Wysocki

**Affiliations:** 1Department of Neonatology, Poznan University of Medical Sciences in Poznan, 60-535 Poznań, Poland; 2Department of Pediatrics, Collegium Medicum, University of Zielona Góra, 65-046 Zielona Góra, Poland; 3Department of Pediatrics, Centre of Postgraduate Medical Education, 01-813 Warsaw, Poland; 4Department of Pharmacoeconomics, Hospital Infection Control Team, Institute of Mother and Child, 01-211 Warsaw, Poland; 5Department of Neonatology and Neonatal Intensive Care, Institute of Mother and Child, 01-211 Warsaw, Poland; 6Sanofi A/S, Vaccines Medical Affairs, DK-2100 København, Denmark; 7Sanofi Sp. z o. o., Vaccines Medical Affairs, 01-211 Warsaw, Poland; pawel.aleksiejuk@sanofi.com; 8PEX Sp. z o. o., 02-796 Warsaw, Poland; asmaga@pzh.gov.pl (A.S.);; 9National Institute of Public Health NIH—National Research Institute, 00-791 Warsaw, Poland; 10Department of Preventive Medicine, Poznan University of Medical Sciences at Poznan, 61-701 Poznan, Poland

**Keywords:** respiratory syncytial virus, RSV, infection, epidemiology, children, prevention

## Abstract

Background: Respiratory syncytial virus (RSV) is one of the leading causes of lower respiratory tract illness and hospitalizations in children aged ≤5 years worldwide. The aim of this study was to characterize the Polish population of patients aged ≤5 years who were hospitalized due to RSV infection, focusing on their clinical and epidemiological characteristics as well as treatment patterns. Methods: This retrospective observational study was conducted between November 2023 and February 2024 in 41 hospitals with pediatric departments across Poland. Data from patients aged ≤5 years admitted due to RSV infection confirmed with antigen test or RT-PCR were collected. The dataset was weighted and extrapolated to allow conclusions applicable to the general population of patients aged 0–5 years hospitalized with RSV infection in Poland. Results: Data from 419 patients were analyzed. Over half (57.4%) were younger than 12 months, 84% were born at term, and 85.8% had no comorbidities. The most frequent manifestations of RSV infections were pneumonia (56.8%), bronchiolitis (35.9%), and bronchitis (12.4%). Viral co-infections were identified in 8% of patients. Regarding treatment, 21.1% of patients required respiratory support, 67.6% received inhaled steroid therapy, 61.5% were treated with antibiotics, 48.1% received beta2-mimetics and anticholinergics, and 44.3% underwent systemic steroid therapy. Conclusions: Our findings confirm that severe RSV primarily affects the youngest children with no comorbidities who do not present high risk conditions. To reduce the overall disease burden, preventive strategies should be offered to all children, not being limited to those in risk groups.

## 1. Introduction

Respiratory syncytial virus (RSV) is one of the leading causes of lower respiratory tract illness (LRTI) and hospitalizations in children aged ≤5 years worldwide [1,2,3,4,5,6]. By the age of 2, nearly all children have been infected with RSV [7,8,9,10,11]. In most cases, RSV infection is mild and self-limiting; however, in some cases it can be more severe, leading to bronchiolitis or pneumonia [7,12,13,14]. RSV causes up to 16 times more hospitalizations and emergency department admissions in children under 5 years than influenza [15,16] and accounts for a substantial proportion of all pediatric hospital admissions. Additionally, RSV-related mortality rates in infants are 10 times higher than those associated with influenza [17].

It has been estimated that in 2019, 33 million cases of RSV-related acute LRTI occurred in children under 5 years old, leading to 3.6 million hospitalizations and over 100,000 deaths [4,18]. Approximately 20% of hospitalized children required admission to an Intensive Care Unit (ICU) [19]. In Poland in 2024, a total of 41,631 cases were recorded, of which 19,080 involved children under 2 years of age [20]. Therefore, RSV places a significant burden on healthcare systems [7,21,22].

Risk factors for a severe course of the disease, aside from young age, include a history of prematurity and underlying medical conditions, such as chronic lung disease, congenital heart disease, and immunodeficiency syndromes [7,23,24]. However, several studies have shown that most children hospitalized due to RSV infection are born at term and were previously healthy [19,25,26].

Given the lack of effective therapies, prevention remains the only way to reduce RSV-related morbidity and mortality. Three monoclonal antibodies, palivizumab, nirsevimab, and clesrovimab, are currently available to prevent a serious course of RSV-related disease in children [27]. Palivizumab requires monthly injections and is restricted to infants born preterm or those with major risk factors [19,28]. Nirsevimab has been approved in the US and EU for use in all newborns and infants during their first RSV season [29]. Clesrovimab has recently been approved in US for prevention of RSV lower respiratory tract disease in neonates and infants born during, or entering, their first RSV season [30]. Nevertheless, although most severe RSV cases occur in children born at term and with no risk factors, current prevention strategies remain focused on at-risk children [21,25,26]. In addition to monoclonal antibodies administered to infants, a bivalent subunit vaccine has been recently approved in the US and UE for pregnant women to prevent RSV-associated LRTI in infants aged <6 months [31,32]. Additional RSV vaccines and monoclonal antibodies are at various stages of clinical development [33]. In Poland, the national policy includes reimbursed immunoprophylaxis against RSV for infants in risk groups, provided as passive immunization with palivizumab for premature newborns and children with underlying medical conditions. In addition, starting from 1 April 2025, RSV vaccination is reimbursed for pregnant women, providing passive protection to their newborns [34,35].

Historically, RSV infections peaked during winter months in each hemisphere [36,37]. However, this predictable pattern was disrupted by nonpharmaceutical interventions implemented during the COVID-19 pandemic [36,38]. After an initial decline of seasonal activity, earlier peaks and higher number of cases were observed out of season [39,40,41]. Recent findings indicate that the seasonal pattern of RSV activity has returned to pre-COVID-19 pandemic trends [42]. The pandemic not only disrupted the temporal pattern of RSV circulation but also affected patient characteristics. Children hospitalized with RSV infection after the COVID-19 outbreak were older and had fewer comorbidities than before the pandemic [43,44].

The incidence and burden of RSV-related hospitalizations in pediatric patients in Poland have been recently described based on National Health Fund (NHF) data for the previous eight seasons [39]. However, no detailed information regarding the patients’ profile, pathway, and treatment patterns are available for Poland. Therefore, our aim was to characterize the population of patients aged ≤5 years hospitalized due to RSV infection in Poland, focusing on their clinical and epidemiological characteristics, as well as treatment patterns.

## 2. Materials and Methods

This retrospective observational study was conducted between November 2023 and February 2024 in hospitals with pediatric departments. As specified in the project framework, no diagnostic or therapeutic interventions or additional specialist consultations were performed. The Bioethics Committee at the Poznan University of Medical Sciences in Poznan (KB-620/23) stated that the study does not meet the criteria of a medical experiment and, in accordance with Polish law and Good Clinical Practice (GCP), does not require approval.

### 2.1. Collected Data

Data were collected using two online questionnaires developed by the authors based on medical documentation. Questionnaire “A” gathered general data on pediatric patients hospitalized for acute respiratory tract infections (ARTI), while questionnaire “B” focused on detailed data for those hospitalized due to RSV infections confirmed by laboratory testing. Data from Questionnaire “A” was used to weight and extrapolate data from Questionnaire “B” to the general population. Questionnaire “B” provided a detailed analysis of patients’ profiles, pathways, and healthcare resource utilization.

### 2.2. Selection of Hospitals

Among all hospitals in Poland, 325 held a contract with the National Health Fund (NHF) for pediatric treatment in 2023. The hospitals were classified into three segments based on contract value, which reflects the number and type of services provided and generally corresponds to the scale of admitted patients. (Appendix A). For feasibility and cost reasons, only hospitals with high and medium contract sizes were included, covering 85% of the total contract volume. These hospitals were divided into 7 macroregions (grouping of voivodeships), resulting in 14 groups (2 contract sizes × 7 regions).

Hospitals were randomly selected using a quota sampling method. Initially, 60 hospitals were divided into high and medium potential centers according to the contract size (43% for high, 57% for medium). For feasibility reasons, the proportion of hospitals was adjusted to 33% high contract size (20 hospitals) and 67% medium contract size (40 hospitals). In total, 41 hospitals were included. Sample sizes for high and medium contract sizes were allocated to 14 groups. Hospitals within each group were randomly selected and assigned to one of two data collection periods, 1 November 2022–15 January 2023 (Period I) or 16 January 2023–31 March 2023 (Period II), to ensure that the sample reflected both phases of the 2022/23 RSV season [39].

Inclusion criteria for the hospitals were as follows:Contains a pediatric department.Admits pediatric patients on an emergency or unplanned basis, commonly referred to as “walk-ins”.If a chosen hospital could not be included due to not meeting the criteria or for other reasons, another hospital in the geographical proximity was selected.

### 2.3. Selection of Doctors

Each hospital was represented by a single doctor. The inclusion criteria for doctors were as follows:Has been working in the pediatric department of the hospital, where patients aged 0–5 years are admitted, for at least 12 months.Provides care of patients hospitalized due to RSV infection.Has access to the medical records of pediatric patients hospitalized due to RSV infections.Is able to refer at least 7 patients to questionnaire B.

### 2.4. Selection of Patients

Codes used for the identification of cases of interest are provided in Appendix B. The inclusion criteria for questionnaire A were as follows:Patients aged ≤5 years on the first day of hospitalization.Hospitalized due to ARTI in a pediatric department during the periods 1 November 2022–15 January 2023 or 16 January 2023–31 March 2023.Every patient in medium contract size centers and every 10th patient in high contract size centers were described, but no more than 30 patients per single center.

Inclusion criteria for patients described in questionnaire B were as follows:Aged ≤5 on the first day of hospitalization.Hospitalized during the analyzed time periods in the pediatric department of the participating doctor.Hospitalized due to RSV infection confirmed with antigen test or RT-PCR.Hospitalization ended in discharge or death.

### 2.5. Data Extrapolation and Weighting

To draw conclusions applicable to the general population of patients aged 0–5 years hospitalized for RSV infection in Poland from 1 November 2022 to 31 March 2023, the data obtained from the research sample were weighted and extrapolated. The weights eliminated an age bias related to the fact that some of the included centers admitted patients in a narrower age range than 0–5 years.

The two-stage weighting and extrapolation process used NHF data on the contract value for pediatric services–hospitalization and a lump sum for the pediatric hospital. The contract value was assumed to be proportional to the number of admitted patients and to data collected in this project regarding the number and characteristics of patients hospitalized due to ARTI during the study period.

## 3. Results

Overall, data from 1052 patients were collected in questionnaire A and from 419 patients in questionnaire B (Table 1). The results presented below show data from the sample described in questionnaire B and extrapolated to the general population of pediatric patients hospitalized due to RSV infections in Poland.

### 3.1. Patient Characteristics

A total of 57.4% of patients were aged under 12 months, 26.8% were between 13 and 24 months old, and the remaining 15.8% were 25–60 months old. The median age was 8.4 months. Overall, 56.4% of patients were female and 95.5% were born from a singleton pregnancy. Additionally, 84% of children were born at term with a median gestational 39 weeks. The median birth weight was 3290 g and 88.3% of children had a birth weight of 2500 g or higher. The median APGAR score was 10; 72.2% of patients had an APGAR score of 10 and 24.5% received a score of 7–9. Only 1.2% of patients received palivizumab and 0.7% had a previous RSV infection. Detailed characteristics are presented in Table 1. Among infants under 12 months of age hospitalized with RSV, 50.5% were born outside the typical RSV season (April–September) (Table 2).

### 3.2. Comorbidities and Risk Factors of RSV Infections

The list of comorbidities and risk factors for RSV infection considered in the study was predefined and is presented in Table 3. A total of 14.2% patients had risk factors or comorbidities. The most common were atopy, frequent respiratory tract infections, and hemodynamically significant heart defects. No risk factors or comorbidities were observed in 90.1% of patients aged <12 months and in 80.1% of patients aged >12 months.

### 3.3. Health State at Admission

At admission, the mean body temperature was 37.2 °C (median 37.0 °C) and the mean O_2_ saturation was 95% (median 95%). Among patients aged ≤12 months, the three most frequent symptoms were rhinitis/runny nose (78%), wheezing (56%), and different-caliber neighs (47%). In case of patients aged >12 months, the most frequent symptoms were rhinitis/runny nose (74%), different-caliber neighs (61%), and fever (60%). In both age groups, viral co-infections were identified in 8% of patients, with the most common ones being influenza A and B, as well as SARS-CoV-2. The most frequent clinical manifestations of RSV infections were pneumonia (56.8%), bronchiolitis (35.9%) and bronchitis (12.4%), and these conditions may have been recorded as co-occurring diagnoses. Detailed data are provided in Appendix A.

### 3.4. Patients’ Pathway and Healthcare Resource Utilization

In 75.6% of cases, patients were referred to the hospital by a doctor. Another 20.9% were transported in emergency mode without a referral (Table 4). The mean length of hospitalization was 5.9 days (median 5 days).

RSV was confirmed by a doctor prior to admission in 25.7% of patients, while in 68.8% of cases, RSV was not confirmed by any diagnostic test before admission (Table 4). For the first test after admission, an antigen/rapid/cassette test was used in 79% of cases, while in the remaining 21% of cases, RT-PCR was applied. The results were positive in 99% of cases. Other commonly performed in-hospital tests included CRP (99.1%), leukocyte count (74.9%), blood gas/acid-base balance tests (74.0%), chest X-ray (52.3%), and procalcitonin (46.5%; Table 4).

A total of 21.1% of patients required respiratory support, including 24.6% of infants aged ≤12 months and 16.3% of children aged 13–60 months. Passive oxygen therapy was the most frequently used method, applied in 98.2% of cases, while high-flow nasal cannulas in 5.4% with similar proportions in both age groups. Non-invasive mechanical ventilation and continuous positive airway pressure were used only in the younger age group (Table 4).

Pharmacotherapy was administered to almost all patients. During the hospital stay inhaled steroid therapy and antibiotics were used most frequently (in 67.6% and 61.5% of cases, respectively), followed by inhalation of beta2-mimetics + anticholinergics (48.1%), systemic steroid therapy (44.3%), inhalation of beta2-mimetics (36.9%), and antipyretics/analgesics/NSAIDs (33.2%).

The medicines prescribed most frequently at discharge were inhaled steroids (54.6% of cases), antibiotics and inhalation of saline (27% each), inhalation of beta2-mimetics + anticholinergics (22.8%), inhalation of hypertonic salt (13.9%), and inhalation of beta2-mimetics (12.0%). Detailed information on pharmacotherapy is included in Appendix A. Other treatment methods frequently applied during hospitalization were fluid therapy (65% of cases), aspiration of nasopharyngeal secretions (32.8%), and physical therapy (22.8%; Table 4).

## 4. Discussion

In our study, 41.8% of patients were under 6 months of age, 57.8% were under 12 months of age, and 84.8% were under 24 months of age. These proportions are consistent with those observed by other authors, who reported a proportion of children under the age of 6 months of between 35% and 67%, a proportion of patients under the age of 12 months of between 50% and 85%, and a proportion of those under the age of 24 months of between 75% and 96% [8,39,45,46,47,48,49]. Our results provide further support for the observation that most RSV-related hospitalizations occur during the first 2 years of life, particularly within the first 6 months.

In our study, most children hospitalized due to RSV infections had no risk factors predisposing to a severe course of the infection. This finding is consistent with observations from other studies reporting that between 63% and 95% of patients are otherwise healthy children [8,11,19,26,39,40,43,46,47,48,49,50,51]. In our study, only 1.2% of patients received palivizumab, which confirms that in Poland—as in other countries—high-risk groups (bronchopulmonary dysplasia, very low birth weight infants) represent a small proportion of the population, and the lower rate of severe RSV infections in these groups is largely due to effective prevention [50]. It has been reported that 93% of children admitted to hospital because of RSV infections were not eligible to receive RSV prophylaxis [52]. Hence, these results suggest that targeting preventive strategies only for children from risk groups can have a limited effect on the total burden of RSV-related infections [50]. Protecting all infants appears to be a more appropriate strategy for reducing the RSV-related burden.

The proportion of RSV patients with comorbidities has been shown to increase with age [8,47]. This trend is also reflected in our results, with comorbidities identified in 9.9% of children aged 0–12 months and 19.9% of children aged 13–60 months.

In our study, the most frequent manifestations of RSV infection were pneumonia, bronchiolitis, and bronchitis. In other studies, bronchiolitis was diagnosed most frequently, in 48–80% of cases, followed by pneumonia (10–29.5%) and bronchitis (28.8%) [8,49,53,54,55]. The observation that the frequency of bronchiolitis decreases and the frequency of pneumonia increases with age [8] is supported by our findings.

In of 68.8% of patients in our study, RSV was not confirmed by any laboratory test prior to admission. Only 25.7% of patients were diagnosed by a doctor in an outpatient (OP) setting. In contrast, 90.6% of patients took an RSV test during hospitalization. A study analyzing testing patterns in four healthcare systems in the US between 2015 and 2023 reported that in pre-COVID-19 seasons approximately 20% of patients were tested for RSV in OP settings, while this proportion was close to 50% in in-patient (IP) settings [56]. Over the years, the proportion of testing increased gradually, with a further increase observed after the onset of the COVID-19 pandemic onset. However, although higher testing was observed between pre- and post-pandemic seasons in all settings, the difference between IP and OP settings remained clearly visible [56].

The low proportion of patients tested in an OP setting may be explained by the perception that testing is unnecessary or too costly. Notably, the proportion of positive results was similar between OP and IP settings [56], suggesting that RSV burden can be underestimated in OP settings [57,58].

In our study, 75.6% of patients were referred to the hospital by a doctor. A similar pattern was described in Germany with 63% of patients seen by a general practitioner prior to admission [8]. It has been also shown that in the youngest age groups, particularly among infants under 6 months, more patients presented directly to the emergency department compared to older patients [8]. This may be partly explained by the faster progression of disease in younger infants and the subtler presentation of early clinical symptoms. All of the patients described in our study received pharmacotherapy. The most prescribed medications were inhaled steroids (68%) and antibiotics (61%). Widespread use of antibiotics has been reported by other authors in Canada (60.5%) [59], Finland (58.5%) [60], Italy (72.4%) [11], Israel and the Netherlands (49%) [61], Germany (43.6%) [8], and Belgium (41.3%) [62]. At the same time, several studies reported a low rate of bacterial superinfections, not exceeding 6% of cases [8,55]. The overuse of antibiotics in RSV infections can be attributed to difficulties in excluding bacterial co-infections in the lower respiratory tract [8] and may lead to unnecessary financial burden, adverse events, and the development of antibiotic resistance [59,62]. In our study, bacterial superinfections occurred only in 0.7% of younger patients and in 4.4% of older patients. These values correspond with other reports [8,52]. This finding could have significant implications for health care providers, as it is evident that the use of antibiotics in children hospitalized with RSV, especially those under 12 months of age, has no benefit and may even be harmful.

On the contrary, antiviral drugs were administered in only 0.9% of cases. The only antiviral drug licensed for severe RSV infection treatment is ribavirin [63]. However, data on both its efficacy and safety in pediatric populations are scarce [63,64]. Moreover, recent research has not demonstrated any beneficial effect of ribavirin [63,65,66,67], and concerns about its potential side effects, inconvenient administration, and expense of the therapy led to its discontinuation from the standard of care for RSV infections [63,64,65].

Our study showed that 67.6% children were treated with inhaled steroids, 48.1% were treated with inhaled beta-mimetics, and 44.3% with systemic steroids. Previous studies and recommendations do not support the use of these therapies in the treatment of RSV among young children [68,69]. Nevertheless, such supportive treatments continue to be used despite their lack of demonstrated benefit [70]. Wider prophylaxis aimed at the population of children within the 1st year of age could significantly limit or even eliminate early exposure of infants to antibiotics and steroids.

An analysis of the length of hospitalization showed that most children were hospitalized for less than 8 days (70.8% of patients aged ≤12 months and 84.4% of those aged >12 months). Our results also indicate that children aged ≤12 months require longer hospital stays more often than older children (30.3% vs. 17% of cases). This finding has a significant meaning for the real estimation of RSV hospitalizations and the health care system burden, which requires at least one bed being occupied in the pediatric department for at least 7 days due to one severe RSV infection of a child below 5 years of age.

To our knowledge, this is the first study describing the clinical and epidemiological characteristics of a broad population of children under 5 years of age hospitalized due to RSV infection in Poland. The study includes a large sample and the results can be generalized to the entire population of children in this age group. It provides detailed patient characteristics, health status, care pathways, and treatment patterns, together with the utilization of medical resources. Another important strength of this study is that all included patients had RSV confirmed by laboratory tests, allowing for precise estimation of incidence.

There are several limitations of this research. Firstly, the NHF data which have been used to estimate the number of cases nationally are data that were collected for administrative purposes and therefore need to be interpreted carefully. Although coverage is complete for all public hospitals in Poland, RSV surveillance and testing has not been systematically implemented in Poland, so time- and region-dependent effects need to be accounted for. Although module B will include only patients with a test-confirmed RSV diagnosis, module A will provide information on the number of patients with an RSV diagnosis not confirmed by a test and the number of patients with an RSV diagnosis without the ICD-10 code identifying RSV, allowing this data to be appropriately related to NHF data. Furthermore, the detailed data collected during the chart review will be based on retrospective data abstraction from medical charts, which could lead to missing data, transcription error, or reporter biases. Measures to address this will include use of a standardized data abstraction form with a data abstraction manual. Another limitation is the definition of the RSV “season”. Although RSV activity in Poland is generally described as spanning from October to April, our study covered only the period from 1 November 2022 to 31 March 2023. We selected this interval because NFZ data indicated that it captured the peak of hospitalizations during the 2022–2023 season. However, omitting the early and late shoulder months may lead to an underestimation of the overall burden of RSV circulation. An additional limitation relates to site selection. The study included only medium- and high-contract-value hospitals, and participating centers were required to contribute a minimum number of cases. Although weighting and extrapolation procedures were applied to adjust for this imbalance and to reproduce the national structure of RSV hospitalizations in children aged 0–5 years, some residual selection bias cannot be fully excluded. Further, information on co-infections was based solely on reported positive results, without details on whether comprehensive respiratory pathogen testing was performed for all patients or which diagnostic methods were used. Therefore, the true prevalence of viral co-infections may be underestimated. Last limitation of this study is the lack of precise operational definitions for certain questionnaire-based variables. Specifically, “frequent, recurrent respiratory tract infections” and “recent respiratory tract infection requiring hospitalization” were recorded as dichotomous items based on clinicians’ judgment, without predefined criteria regarding the number of infections or a clearly specified time frame for hospitalization. This may limit the interpretability and reproducibility of these variables, and future studies should apply standardized definitions and explicit temporal criteria.

## 5. Conclusions

Our data confirm that severe RSV mostly affects the youngest children with no comorbidities, who typically do not qualify for RSV prophylaxis with currently available monoclonal antibodies. Consequently, to reduce overall disease burden, preventive strategies should be provided to all children, not only those in risk groups. The high frequency of antibiotic use also suggests that widespread preventive strategies could be a better way of managing RSV than treatment entailing adverse events and potential drug resistance, despite the low rate of confirmed bacterial infections.

## Figures and Tables

**Table 1 viruses-18-00060-t001:** Characteristics of patients ≤ 5 years hospitalized with RSV; weighted % (*n*).

	Whole Population
Age at onset of hospitalization (months) *n* = 419	
0–1	3.8% (19)
2–3	18.6% (93)
4–6	18.7% (101)
7–12	16.3% (84)
13–24	26.8% (64)
25–60	15.8% (58)
Gender *n* = 419	
Male	43.6% (239)
Female	56.4% (180)
Pregnancy type *n* = 383 *	
Singleton	95.5% (362)
Multiple	4.5% (21)
Patient’s gestational age (weeks) *n* = 300 *	
<29	1.2% (4)
29–31	1.1% (4)
32–36	14.2% (41)
≥37	83.5% (251)
Birth weight (g) *n* = 310 *	
<750	0.3% (1)
750–999	0.4% (1)
1000–1499	1.1% (4)
1500–2499	9.9% (32)
2500–3999	77.7% (241)
≥4000	10.6% (31)
APGAR score *n* = 320 *	
2	1.2% (3)
4	0.5% (2)
5	0.6% (2)
6	1.0% (3)
7	4.4% (13)
8	4.2% (13)
9	15.8% (53)
10	72.2% (231)
Previous use of palivizumab *n* = 361 *	
Yes	1.2% (5)
No	98.8% (356)
Previous RSV infections *n* = 313 *	
Yes	0.7% (2)
No	99.3% (311)

*—smaller population due to missing data.

**Table 2 viruses-18-00060-t002:** Month of birth of patients aged ≤12 months.

Month	Patients Aged ≤12 Months (*n* = 289)
January	6.9%
February	3.0%
March	2.9%
April	4.8%
May	5.4%
June	7.2%
July	10.5%
August	10.9%
September	11.8%
October	16.8%
November	11.8%
December	8.0%

**Table 3 viruses-18-00060-t003:** Risk factors in the whole population and subgroups by age; weighted % (*n*).

Risk Factors/Comorbidities	All Patients (*n* = 406) *	Patients Aged ≤12 Months (*n* = 289)	Patients Aged >12 Months (*n* = 117)	*p*-Value
Atopy	4.2% (15)	1.5% (5)	7.8% (10)	<0.05
Frequent, recurrent respiratory tract infections	4.0% (14)	0.7% (2)	8.6% (12)	<0.05
Hemodynamically significant heart defects	2.5% (8)	1.2% (3)	4.4% (5)	<0.05
Recent respiratory infection requiring hospitalization	2.2% (11)	1.9% (6)	2.6% (5)	ns
Asthma	1.7% (5)	0% (0)	4.1% (5)	<0.05
Bronchopulmonary dysplasia	1.5% (7)	1.6% (5)	1.4% (2)	ns
Neurological disorders with hypotonia	1.1% (3)	0.7% (2)	1.7% (1)	ns
Cyanotic heart defects	0.9% (3)	0.8% (2)	1.1% (1)	ns
Cerebral palsy	0.2% (1)	0%	0.5% (1)	ns
Cystic fibrosis	0%	0%	0%	ns
Down syndrome	0%	0%	0%	ns
Immunodeficiency	0%	0%	0%	ns
Moderate to severe secondary pulmonary hypertension	0%	0%	0%	ns
Overt heart failure persisting despite drug treatment	0%	0%	0%	ns
Percutaneous arterial oxygenation below 90%	0%	0%	0%	ns
Planned cardiac surgery	0%	0%	0%	ns
Congenital malformations of the circulatory system other than those mentioned above	2.1% (11)	3.4% (10)	0.5% (1)	<0.05
Congenital malformations of the respiratory system other than those mentioned above	0.2% (1)	0.3% (1)	0%	ns
A chronic respiratory disease, different from those mentioned above, originating in the perinatal period.	0.2% (1)	0%	0.5% (1)	ns
A chronic disease other than the one mentioned above	5.0% (21)	3.8% (12)	6.5% (9)	ns
None of the above	85.8% (347)	90.1% (259)	80.1% (88)	<0.05

*—smaller population due to missing data; ns—not significant.

**Table 4 viruses-18-00060-t004:** Patient pathway and healthcare resources utilization; weighted % (*n*).

	Whole Population
**Duration of hospitalization *n* = 419 (days by date of admission and discharge)**	
0	0.7% (4)
1	1.9% (6)
2	7.7% (29)
3	11.6% (46)
4	13.3% (50)
5	16.2% (62)
6	13.1% (52)
7	11.2% (54)
8–14	22.8% (108)
15–21	1.0% (5)
22–28	0.6% (3)
**Mode of admission to hospitalization (*n* = 414) ***	
Referral by a doctor (AOS, outpatient clinic)	75.6% (302)
Emergency mode without referral (ED, ambulance)	20.9% (97)
Transfer from another hospital	3.3% (14)
Transfer from another ward of the same hospital (the original reason for hospitalization was not a respiratory infection)	0.2% (1)
**Confirmation of RSV prior to admission to hospital (*n* = 381) ***	
No RSV confirmation with a test performed	68.8% (260)
RSV confirmed by a test performed by a doctor	25.7% (99)
RSV confirmed by a home test performed by the patient’s parents/caregivers	3.6% (13)
RSV confirmed by a test performed differently than in the above.	1.8% (9)
**Performing a test for RSV as part of hospitalization (*n* = 418) ***	
Yes	90.6% (379)
No	9.4% (39)
**Diagnostic/control tests performed on the patient during hospitalization (*n* = 419)**	
CRP	99.1% (414)
Leukocyte count	74.9% (314)
Blood gas/acid-base balance tests	74.0% (313)
Chest X-ray	52.3% (214)
PCT (procalcitonin)	46.5% (198)
Lung ultrasound	31.8% (136)
Serum total IgE	6.6% (25)
OB	4.4% (18)
Computed tomography of the lungs	0.2% (1)
Other tests	24.9% (94)
**Treatment methods for RSV infection applied during hospitalization (without respiratory support) (*n* = 419)**	
Pharmacotherapy	99.7% (417)
Fluid therapy	64.9% (266)
Aspiration of nasopharyngeal secretions	32.8% (150)
Physical therapy	22.8% (109)
Nasogastric or naso-gastroduodenal tube feeding	0.6% (3)
Non-pharmacological treatment other than those mentioned above	0.6% (3)
**Respiratory support methods applied to the patient during hospitalization; weighted % (*n*) *n* = 103**	
Passive oxygen therapy	98.2% (101)
High Flow Nasal Cannulas (HFNC)	5.4% (6)
Non-invasive mechanical ventilation (NIV)	1.9% (2)
Continuous Positive Airway Pressure (CPAP)	0.9% (1)

*—smaller population due to missing data.

## Data Availability

Data are available from the authors upon reasonable request.

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
