# Peer review of "Clinical Burden and Healthcare Utilization Associated with Hospitalizations of RSV-Infected Polish Children During the 2022/23 Season"

_viruses, 2025, doi:10.3390/v18010060_

Round 1
Reviewer 1 Report
Comments and Suggestions for Authors
It is an interesting article offering results regarding RSV from Eastern European part data that are very welcome. However some question do come up. Were there any differences in regard of the answers when using the questionnaire during different seasons and/or months?
What was the vaccine coverage against RSV if any? Could the authors also comment on the national policy regarding RSV vaccine both at infants, premature or pregnant individuals?
What is thee national birth cohort and what is the incidence and prevalence of the disease in the country?
Did any of the infants and toddlers with chronic conditions receive any prior immunization against RSV?
Author Response
It is an interesting article offering results regarding RSV from Eastern European part data that are very welcome. However some question do come up.
Authors: Thank you for your positive assessment of our work and for highlighting the value of Eastern European RSV data. We appreciate your feedback and will address all questions and comments in detail to further strengthen the manuscript.
Were there any differences in regard of the answers when using the questionnaire during different seasons and/or months?
Authors: Thank you for this question. We do not have such analyses or results, as comparing patient subpopulations between the two periods was not the aim of the study. The purpose of dividing the season into two consecutive 10-week intervals was to achieve the most even distribution of patients across the entire observation period, ensuring the greatest possible representativeness of the sample.
What was the vaccine coverage against RSV if any? Could the authors also comment on the national policy regarding RSV vaccine both at infants, premature or pregnant individuals?
Authors: Thank you for this comment. In our study, previous use of palivizumab was reported in 5 patients (1.2%), and these data are presented in Table 1. We have also added information in the Introduction regarding the national policy on RSV vaccination in infants, premature newborns, and pregnant individuals.
What is the national birth cohort and what is the incidence and prevalence of the disease in the country?
Authors: Thank you for this comment. We have added some statistics and numerical data to the Introduction.
Did any of the infants and toddlers with chronic conditions receive any prior immunization against RSV?
Authors: Thank you for this comment. We note in our manuscript that five individuals received palivizumab immunization. Due to this very small number, it is not possible to perform a meaningful analysis of how many patients with chronic conditions were included in this group.
Reviewer 2 Report
Comments and Suggestions for Authors
1. Only two periods (1 November 2022 to 31 March 2023) were sampled, whereas RSV circulation in Poland typically extends from October to April; omitting the shoulder periods may lead to systematic underestimation.
In addition, only medium- and high-contract-value centers were included, and each site was required to have one physician who could “refer ≥7 cases,” which may bias the sample toward departments with higher patient volume, more complete documentation, and greater willingness to participate.
2. The Methods section states that Table B includes only laboratory-confirmed RSV hospitalizations, yet the Results report that 9.4% of patients had no RSV testing performed during hospitalization.
3. The appendix lists ICD-10 codes, but the primary dataset was obtained through case-by-case chart extraction rather than a secondary database query.
4. Although the study is framed as an assessment of clinical burden and healthcare resource utilization, key outcomes such as ICU admission, mechanical ventilation, mortality, readmissions, and direct medical costs or cost per hospital day were not systematically reported.
5. The reported proportions—pneumonia 56.8%, bronchiolitis 35.9%, bronchitis 12.4%—clearly reflect multiple coexisting diagnoses, but the text does not explicitly state that these categories are non-mutually exclusive.
6. Antibiotic and systemic corticosteroid use was high, yet the manuscript does not systematically relate these prescribing patterns to the very low documented rates of bacterial co-infection.
Author Response
Only two periods (1 November 2022 to 31 March 2023) were sampled, whereas RSV circulation in Poland typically extends from October to April; omitting the shoulder periods may lead to systematic underestimation.
Authors: Thank you for this comment. Based on the National Health Fund (NFZ) data, we observed that this particular season was shifted. We therefore selected the range of months that maximized the number of hospitalizations for that season. In general, the concept of a “season” appears somewhat conventional in this context, as RSV-related hospitalizations occur throughout the entire year, including outside the typical “season.”
Below is the relevant excerpt of hospitalization data obtained from the NFZ:
2022_01 – 603
2022_02 – 150
2022_03 – 136
2022_04 – 92
2022_05 – 104
2022_06 – 74
2022_07 – 47
2022_08 – 14
2022_09 – 89
2022_10 – 572
2022_11 – 4,319
2022_12 – 8,940
2023_01 – 4,179
2023_02 – 1,535
2023_03 – 670
2023_04 – 241
2023_05 – 93
2023_06 – 15
We have nevertheless added this point as a limitation in the Discussion section.
In addition, only medium- and high-contract-value centers were included, and each site was required to have one physician who could “refer ≥7 cases,” which may bias the sample toward departments with higher patient volume, more complete documentation, and greater willingness to participate.
Authors: To formulate conclusions based on the study results referring to the total population of patients aged 0-5 hospitalised due to RSV over the 01.11.2022 - 31.03.2023 period in Poland, the data obtained from the research sample was weighted and extrapolated. Weights have been designed to adjust a disproportionate sample due to the hospital segment expressed by means of the value of the contract with the NHF and reproduction of the age structure of patients hospitalised due to RSV eliminating the disturbance resulting from incorporating in the research HCPs working at wards admitting patients from a narrower group than 0-5 years of age.
A two-stage weighting and extrapolation process was employed, utilising the NHF data on the value of contracts for the ‘paediatrics – hospitalisation’ and ‘PSZ lump sum – paediatric hospital’ services in 2023, which according to authors’ assumption are proportionate to the number of patients managed, as well as the data gathered in the Economedica RSV study – Form (A) register 2 and 3 (the number and profile of patients hospitalised due to acute respiratory tract infections during the period under review). The weighting and extrapolation process comprised of the following steps:
Estimating the number of hospitalisations due to acute respiratory tract infections at the age of 0-5 in Poland – split into age groups.
Hospital segmentation according to the contract value into the following three groups:
[P1] above PLN 8.2 million (49 hospitals accounting for 38% of the total value of the contract for this service)
[P2] from PLN 2.97 million to PLN 8.2 million (190 hospitals accounting for 51% of the total value of the contract for this service)
[P3] EXCLUDED: below PLN 2.97 million (86 hospitals accounting for 11% of the total value of the contract for this service)
Exclusion of hospitals at which a HCP taking part in the research worked at a ward managing children from a narrower age group than 0-5 or at which a HCP erroneously described only RSV patients in Form (A) register 3
For each of two periods under examination (I, II) calculation of the projection coefficient according to contract value (quotient of the total value of the contracts for hospitals from a given segment and the total value of contracts for hospitals in the sample) and the coefficient adjustment resulting from the value of the contract for hospitals from a segment skipped in the research (1.12 multiplier).
Also, we have added explanation to the limitations.
The Methods section states that Table B includes only laboratory-confirmed RSV hospitalizations, yet the Results report that 9.4% of patients had no RSV testing performed during hospitalization.
Authors: Thank you for this question. Eligibility for inclusion in the study required a “yes” answer to the item: “R1. Was RSV diagnosed in this patient during the analysed hospitalization based on an antigen test or RT-PCR?” Two additional questions in the form also captured information on testing performed before admission (C2) and on whether an RSV test was performed during hospitalization (D1).
Therefore, if a patient did not undergo RSV testing during the hospitalization itself, this implicitly means that RSV had been confirmed before admission. Such patients were still eligible, as confirmation prior to hospitalization fulfilled the study’s inclusion criterion.
The appendix lists ICD-10 codes, but the primary dataset was obtained through case-by-case chart extraction rather than a secondary database query.
Authors: ICD-10 codes identifying patients with ARTI and RSV were used only for selecting patients for Form A (where an RSV patient did not necessarily need test confirmation). In contrast, for Form B no ICD-10 codes related to RSV were used for patient selection, because only test-confirmed RSV cases were included, making such coding unnecessary. However, in Form B we also collected information on ICD-10 codes assigned as the final diagnosis for the analysed hospitalization.
Although the study is framed as an assessment of clinical burden and healthcare resource utilization, key outcomes such as ICU admission, mechanical ventilation, mortality, readmissions, and direct medical costs or cost per hospital day were not systematically reported.
Authors: Thank you for this comment. Some of the outcomes you mention are presented in Section 3.4 of the Results. Other elements, including direct medical costs, were not included because a cost or economic analysis was not an objective of this study. The focus of the manuscript was on describing the clinical course and healthcare resource utilization in terms of hospitalization characteristics, rather than providing a full health-economic evaluation.
The reported proportions—pneumonia 56.8%, bronchiolitis 35.9%, bronchitis 12.4%—clearly reflect multiple coexisting diagnoses, but the text does not explicitly state that these categories are non-mutually exclusive.
Authors: Thank you for this point. We have clarified that these conditions may have been recorded as co-occurring diagnoses.
Antibiotic and systemic corticosteroid use was high, yet the manuscript does not systematically relate these prescribing patterns to the very low documented rates of bacterial co-infection.
Authors: We agree with this observation. Antibiotics were clearly overused in relation to the very low documented rates of bacterial co-infection. We have added an explicit note highlighting this point in the Conclusions section.
Reviewer 3 Report
Comments and Suggestions for Authors
I would like to thank you for giving me the opportunity to review your valuable paper.
This study holds clinical value as the first retrospective observational study to describe the current clinical picture of pediatric RSV infection in Poland on a relatively large scale.
I was particularly surprised by how significantly the inpatient treatment methods diverged from international standard practices. The frequency of antibiotic use, inhaled steroid therapy and systemic steroid administration is truly astonishing.  I am also surprised by the infrequency of nasal suctioning, which is an important physical therapy procedure for neonates and early infants with RSV.
There are several concerns that require your consideration.
The definitions for “history of frequent respiratory infections” and “history of respiratory infections requiring hospitalization” in Table 3 are not provided (line191).
Regarding co-infections with RSV, while influenza A, B, and SARS-CoV-2 are mentioned, it is necessary to specify the extent of testing performed on the subjects. Does this mean that all cases were examined using comprehensive PCR methods for respiratory viruses? The type of virus detection test method should be mentioned (line199-200). Incidentally, in our single-center study (2022-2024), RV/EV was overwhelmingly the most common co-infection in more than 300 RSV patients.
Additional discussion is needed regarding the study's limitations, specifically how the retrospective nature of the multi-center study and the evaluation of data consistency and authenticity were addressed.
The conclusion states that the frequency of pediatric hospitalizations due to RSV infection following the COVID-19 pandemic—the period under study—was higher than predicted by prior epidemiological data, but it does not provide the basis for this assertion. Furthermore, it is unclear whether the increase in hospitalized patients stems from a rise in the number of infected individuals or an increase in the hospitalization rate per patient. I consider these points to be of critical importance from the epidemiological point of view (line331-332).
Author Response
I would like to thank you for giving me the opportunity to review your valuable paper.
This study holds clinical value as the first retrospective observational study to describe the current clinical picture of pediatric RSV infection in Poland on a relatively large scale.
I was particularly surprised by how significantly the inpatient treatment methods diverged from international standard practices. The frequency of antibiotic use, inhaled steroid therapy and systemic steroid administration is truly astonishing.  I am also surprised by the infrequency of nasal suctioning, which is an important physical therapy procedure for neonates and early infants with RSV.
There are several concerns that require your consideration.
Authors: Thank you very much for your thoughtful and supportive remarks. We appreciate your review and the perspectives you provided. Please see our detailed responses to your comments, where we address the issues you raised.
The definitions for “history of frequent respiratory infections” and “history of respiratory infections requiring hospitalization” in Table 3 are not provided (line191).
Authors: These two terms were interpreted literally in the study: (1) frequent or recurrent respiratory infections, and (2) a recent respiratory infection that required hospitalization.
Regarding co-infections with RSV, while influenza A, B, and SARS-CoV-2 are mentioned, it is necessary to specify the extent of testing performed on the subjects. Does this mean that all cases were examined using comprehensive PCR methods for respiratory viruses? The type of virus detection test method should be mentioned (line199-200). Incidentally, in our single-center study (2022-2024), RV/EV was overwhelmingly the most common co-infection in more than 300 RSV patients.
Authors: Thank you for this important point. Unfortunately, the structure of our questionnaire does not allow us to determine whether all patients were systematically tested for pathogens other than RSV, nor which specific diagnostic methods (e.g., PCR panels) were used. The survey item related to co-infections was phrased as follows:
“D4. During the analysed hospitalization, did the patient have a positive test result for any of the pathogens listed below (in addition to or independently of RSV testing)?”
The list included adenovirus, multiple human coronaviruses (229E, HKU1, NL63, OC43, MERS-CoV, SARS-CoV-2), human metapneumovirus, rhinovirus/enterovirus, influenza A and B subtypes, parainfluenza viruses, and an option for “other” or “none.”
Because the question only asked about positive results, and did not ask whether testing was performed for each pathogen or what testing platform was used, we cannot infer whether comprehensive respiratory virus PCR panels were applied to all subjects. This is an important limitation of the dataset, and we now emphasise this point in the manuscript.
Additional discussion is needed regarding the study's limitations, specifically how the retrospective nature of the multi-center study and the evaluation of data consistency and authenticity were addressed.
Authors: Thank you for this comment. We have substantially expanded the limitations section in our Discussion.
The conclusion states that the frequency of pediatric hospitalizations due to RSV infection following the COVID-19 pandemic—the period under study—was higher than predicted by prior epidemiological data, but it does not provide the basis for this assertion. Furthermore, it is unclear whether the increase in hospitalized patients stems from a rise in the number of infected individuals or an increase in the hospitalization rate per patient. I consider these points to be of critical importance from the epidemiological point of view (line331-332).
Authors: Thank you. We agree that this statement could be misleading for the reader, so we decided to remove it. We appreciate you bringing this to our attention.
Round 2
Reviewer 2 Report
Comments and Suggestions for Authors
none
Author Response
Thank you for the review and for accepting the manuscript.
Reviewer 3 Report
Comments and Suggestions for Authors
I would like to thank you for giving me the opportunity to review your revised paper.
I appreciate that you have generally responded sincerely to the reviewers' questions.
Regarding the first question, it seems you haven't quite grasped it, so I'll explain it again.
There is no clear medical definition for the term “frequent infections.”
Instead, the following are often used as clinical indicators to assess the possibility of immunodeficiency.
Pneumonia two or more times a year
Middle ear infections four or more times a year
Two or more serious sinus infections within 1 year, and so on.
In this study, I believe it is necessary to clarify how “frequent infections” were defined.
It is also necessary to clarify what period of time the term “recent hospitalization history” refers to. Within the last 3 months? Within the last 6 months? Within the last year? Within the last 2 years?
If the above definitions are unclear, I imagine it would be difficult to answer the “questionnaire.” What do you think?
Author Response
Thank you for this important and clarifying comment.
You are correct that there is no universally accepted medical definition of the term “frequent infections,” and that in clinical practice this concept is usually operationalized using specific criteria (e.g. number of pneumonias, otitis media episodes, or sinus infections within a defined time frame).
In our study, however, the data were collected using a predefined questionnaire, in which “frequent, recurrent respiratory tract infections” and “recent respiratory tract infection requiring hospitalization” were included as separate items among comorbidities and risk factors. These items were assessed in a dichotomous manner (yes/no), based on the treating physician’s clinical judgment at the time of completing the questionnaire. Unfortunately, the questionnaire did not include more detailed operational definitions, such as the exact number of infections or a strictly defined time window.
Similarly, the term “recent hospitalization history” referred to a respiratory tract infection requiring hospitalization in the recent period, but no specific time frame (e.g. 3, 6, or 12 months) was predefined in the questionnaire.
We fully agree that the lack of precise definitions is a limitation of the study and may affect the interpretability and reproducibility of these variables. This limitation has now been explicitly acknowledged in the revised manuscript.
Despite this limitation, we believe that these variables still reflect real-world clinical assessment, as they were reported by physicians familiar with the patients’ medical histories. Nevertheless, future studies should apply standardized and clearly defined criteria for “frequent infections” and specify explicit time frames for hospitalization history to improve data precision.